# Behavioral sensitization and tolerance induced by repeated treatment with ketamine enantiomers in male Wistar rats

**Kristian Elersič**[ID]*, **Anamarija Banjac, Marko Živin, Maja Zorović**[ID]

Brain Research Lab, Institute of Pathophysiology, Faculty of Medicine, University of Ljubljana, Ljubljana, Slovenia

* kristian.elersic@mf.uni-lj.si

## Abstract

Ketamine has gained significant attention as a fast-acting antidepressant. However, ketamine is also associated with undesirable side effects. In our preclinical study, we explored the behavioral effects of ketamine enantiomers at subanesthetic doses. During repeated intermittent treatment, we examined locomotor stimulation and sensitization, ataxia, and expression of natural behaviors (grooming and rearing). Male Wistar rats were subcutaneously treated repeatedly with either 5 mg/kg of R-ketamine or S-ketamine, 15 mg/kg of R-ketamine, S-ketamine or racemic ketamine, 30 mg/kg of racemic ketamine or saline every third day for three weeks (seven treatments overall). After the first treatment, only 15 mg/kg of S-ketamine induced locomotor stimulation, and both 15 mg/kg of S-ketamine and 30 mg/kg of racemic ketamine induced ataxia. Upon repeated administration, doses of 15 mg/kg of R-ketamine, S-ketamine, and racemic ketamine, as well as 30 mg/kg of racemic ketamine, stimulated locomotion. 15 mg/kg of R-ketamine, S-ketamine, and racemic ketamine additionally resulted in locomotor sensitization. The last administration of 15 mg/kg of S-ketamine, 15 mg/kg of racemic ketamine, and 30 mg/kg of racemic ketamine resulted in ataxia. In the case of 15 mg/kg of S-ketamine, ataxic effects were significantly weaker in comparison to the effects from the first administration, indicating tolerance. Natural behaviors were attenuated after 5 and 15 mg/kg of S-ketamine and 15 and 30 mg/kg of racemic ketamine. Neither of the R-ketamine doses produced such an effect. We conclude that S-ketamine has a stronger behavioral effect than R-ketamine.

## 1. Introduction

Ketamine, a fast-acting antidepressant, has shown its value as a treatment option for patients suffering from treatment-resistant depression and suicidality [1]. However, ketamine is also associated with undesirable side effects. Subanesthetic antidepressant doses can acutely induce psychotic symptoms and changes in affective states [2, 3]. Moreover, repeated administration of ketamine is necessary for prolonged antidepressant effects [4], raising concerns about its safety [5, 6]. To address these concerns, it is crucial to understand how ketamine works. More

**Data Availability Statement:** The data supporting this study's findings are available on Figshare Elersič, Kristian; Banjac, Anamarija; Živin, Marko; Zorović, Maja (2023). (DATA) Differential behavioral effects of repeated treatment with

ketamine enantiomers in male Wistar rats. figshare. Dataset. https://doi.org/10.6084/m9.figshare.23898696.v2.

**Funding:** This study was funded by the Slovenian Research and Innovation Agency (grant P3-0171 and the young researcher grant to Kristian Eleršič). Both funds were acquired by Marko Živin. The funders had no role in study design, data collection and analysis, decision to publish, or preparation of the manuscript.

**Competing interests:** The authors have declared that no competing interests exist.

specifically, how each enantiomer that comprises racemic ketamine, either R- or S-ketamine, contributes to antidepressant efficacy and side effect profiles [7].

Preclinical studies have reported fewer unwanted behavioral effects [8–12] and greater antidepressant efficacy of R-ketamine [10, 11, 13, 14]. However, the main problem is that such preclinical studies cover only short-term treatments, thus not mimicking real-life antidepressant therapy. Therefore, our study aimed to assess divergent behavioral effects of ketamine enantiomers in repeated intermittent treatment. We administered 5 mg/kg of R-ketamine (R5), 5 mg/kg of S-ketamine (S5), 15 mg/kg of R-ketamine (R15), 15 mg/kg of S-ketamine (S15), 15 mg/kg of racemic ketamine (RS15), 30 mg/kg of racemic ketamine (RS30) or saline (Sal) every third day for three weeks (seven treatments overall).

We focused on behaviors that are usually investigated as unwanted side effects of ketamine: locomotor stimulation, locomotor sensitization, and ataxia. In addition, we checked for alterations in natural behaviors, such as grooming and rearing. Locomotor stimulation is correlated with the activation of the mesolimbic dopaminergic system [15], which is implicated in addiction [16]. Locomotor sensitization is likewise considered a critical factor in addiction development and is associated with changes in the mesolimbic dopaminergic system [17–19]. It refers to an amplified behavioral and physiological response following repeated exposure to a substance. Ataxia is characterized by a disturbance of the righting reflex and discoordination and was used as a preclinical marker of the hypnotic properties of ketamine [12, 20]. Lastly, we analyzed the change in natural behaviors to additionally assess the behavioral effects of ketamine enantiomers.

Previous studies have reported locomotor stimulation after administering racemic ketamine [18, 21–25] and S-ketamine [8, 10, 11, 14]. R-ketamine provided mixed results, either stimulating [8, 14] or not stimulating locomotion [10, 11]. Locomotor sensitization was reported after repeated treatment with racemic ketamine [18, 22–24, 26, 27] and S-ketamine, but not R-ketamine [8]. These findings suggest that S-ketamine has stronger locomotor effects than R-ketamine and may be primarily responsible for the locomotor effects of racemic ketamine. However, the locomotor effects of ketamine enantiomers are understudied, especially in long-term treatment protocols, indicating a need for further investigation.

Studies on racemic ketamine in rodents have demonstrated its ability to induce ataxia [21, 25, 26, 28, 29] and decrease natural behaviors [28–30]. After repeated treatment, no tolerance to the ataxic effects was found [26]. Furthermore, ataxia is stronger after treatment with S-ketamine in comparison with R-ketamine [12]. Unfortunately, no study has focused on the effects of repeated treatment with ketamine enantiomers on ataxia and natural behaviors.

Importantly, our study employed the subcutaneous (s.c.) route of administration, in contrast to the more commonly used intraperitoneal (i.p.) method. We chose the s.c. route due to the scarcity of ketamine studies using the s.c. route. Furthermore, s.c. administration has been shown to yield ketamine exposure levels comparable to those achieved via intravenous infusion (i.v.i.) route [31], which is frequently used in clinical settings. This factor gives translational validity to s.c. administration route. Additionally, clinical research is exploring s.c. ketamine as a therapeutic option [32–34], underscoring the relevance of its preclinical evaluation.

To summarize, our study aimed to explore the divergent effects of repeated treatment with ketamine enantiomers and racemic ketamine, focusing on locomotor stimulation, locomotor sensitization, ataxia, and changes in natural behaviors. We hypothesized that R-ketamine will exhibit less pronounced behavioral effects than S-ketamine, as reported previously in other studies using shorter protocols [8, 10–12].

## 2. Material and methods

### 2.1 Animals

Fifty-five male Wistar rats (10–12 weeks old, 300–380 g) were divided into seven groups: R5, receiving 5 mg/kg of R-ketamine ($n = 7$), S5, receiving 5 mg/kg of S-ketamine ($n = 7$), R15, receiving 15 mg/kg of R-ketamine (LOC1: $n = 8$, LOC2: $n = 7$; one animal was excluded due to high-stress response to treatment protocol), S15, receiving 15 mg/kg of S-ketamine ($n = 8$), RS15, receiving 15 mg/kg of racemic ketamine ($n = 6$), RS30, receiving 30 mg/kg of racemic ketamine (LOC1: $n = 6$, LOC2: $n = 5$; one animal was excluded due to high-stress response to treatment protocol) and Sal, receiving saline ($n = 12$). Groups with variable numerus are a result of a two-step protocol carried out on two cohorts of animals. We used the first cohort (18 animals) to test S15, R15, and saline ($n = 6$ per group). We used the second cohort (37 animals) for groups R5 ($n = 7$), S5 ($n = 7$), RS15 ($n = 6$), and RS30 ($n = 6$). We used additional animals for the same treatment groups as in the first cohort to ensure that animals from both cohorts were included: R15 ($n = 2$), S15 ($n = 2$), and saline ($n = 6$).

Rats were housed in polycarbonate cages (Ehret IV, Mahlberg, Germany, floor area 1825 cm$^2$), three per cage, with sterilized bedding material (Lignocel ¾ and Rehofix, JRS, Rosenberg, Germany), sterilized cellulose towels, and sterilized carton tunnels for enrichment. Housing cages were in a temperature-controlled colony room at 22–24°C (relative humidity 35–60%) under a reversed 12 h light/12 h dark cycle (lights off at 11 a.m.). Animals had free access to autoclaved water and a maintenance rodent diet (1320 Altromin, Lage, Germany). We handled rats for two weeks before the start of the experimental protocol. Both male and female experimenters carried out the study. The male researcher performed all behavioral tests, and the female researcher completed all subcutaneous injections. At the end of the study, we humanely euthanized animals with gradual carbon dioxide release and decapitation. All efforts were made to minimize the suffering of animals. The study was approved by the Administration for Food Safety, Veterinary Sector and Plant Protection of the Republic of Slovenia (Reference Number U34401-34/2020/15 and U34401-2/2022/9), and animals were treated according to Directive 2010/63/EU of the European Union and the National Veterinary Institute Guide for the Care and Use of Laboratory Animals.

### 2.2 Chemicals

R-ketamine hydrochloride (Cat. No. 6751, CAS No. 1867-66-9, Tocris, UK) and S-ketamine hydrochloride (Cat. No.4379, CAS No. 33643-47-9, Tocris, UK) were dissolved in sterile 0.9% saline. For racemic ketamine, we mixed equal quantities of R-ketamine and S-ketamine. Drugs were injected subcutaneously (s.c.) into the neck or hip region at 2 ml/kg. We chose a 15 mg/kg dosage, aligning with the range of S-ketamine doses (10 mg/kg and 20 mg/kg) previously associated with adverse effects [8–10, 14]. To assess the impact of lower doses, we administered 5 mg/kg of ketamine enantiomers, the dose at which racemic ketamine still exhibits antidepressant effects when applied s.c. [35]. Lastly, we administered 30 mg/kg of racemic ketamine to achieve a combined dose of 15 mg/kg of R-ketamine and S-ketamine. All ketamine doses are expressed as the weight of the salt (hydrochloride). To simulate the induction phase of S-ketamine antidepressant treatment [1, 36], the animals were treated every third day for three weeks (seven treatments altogether). We opted for every third day in order to achieve a less variable treatment protocol compared to biweekly treatments that are usually used in a clinical setting.

### 2.3 Behavioral tests

We started behavioral testing at the beginning of the dark phase (11 a.m.). After acclimation in the experimental room for 10 minutes, we removed one animal at a time from the home cage

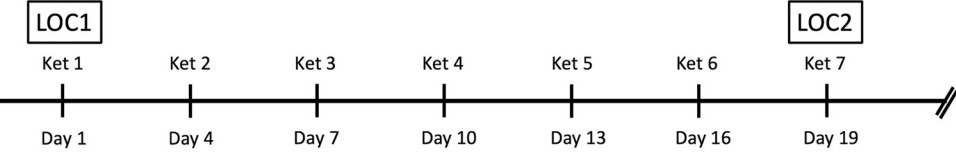

**Fig 1. Study protocol.** LOC (locomotor activity test), Ket (ketamine).

and put it into the testing arena. To minimize the carryover effects of stress, we temporarily housed each tested animal in a holding cage after the test. We returned it to the home cage only after removing the next animal.

To investigate the behavioral effects of ketamine enantiomers and racemic ketamine at different doses, we conducted two locomotor activity tests (LOC): LOC1 on day 1 following the first treatment dose and LOC2 on day 19 following the last treatment dose (Fig 1). We used a black open field arena (1 m × 1 m x 0.5 m) illuminated with a red light (6 lux). The LOC protocol consisted of two steps. First, we measured the basal activity for 20 minutes. Next, we administered treatment solution s.c. and returned the animal to the arena for 60 minutes. We recorded the animals with a camera placed centrally above the arena. To track the locomotor activity, we used EthoVision XT 17 (Noldus, Wageningen, NL). We manually analyzed the first 15 minutes of the recording to assess ataxia and natural behaviors. Ataxia was assessed as the relative number of falls (the number of falls per distance) during the first 15 minutes after the administration. We opted for this method to account for the probability of falls in relation to locomotor stimulation; if an animal moves more, it is more likely to fall. In addition to the first 15 minutes, we exclusively evaluated ataxia for RS30 during the last 15 minutes (RS30-L) of the locomotion test since, in this group, ataxia was most pronounced during that period. Lastly, natural behaviors were evaluated as the time spent grooming and rearing. All manual evaluations were carried out blinded to treatment groups and LOC, with the exception of a post-hoc analysis of ataxic behaviors in the last 15 minutes after the RS30 treatment.

## 2.4 Statistical analysis

Initially, we checked the assumptions of normality using the Shapiro-Wilk test and homoscedasticity with Bartlett's (in case of normal distribution) or Levene's (in case of non-normal distribution) tests. Our data is mainly heteroscedastic and non-normally distributed. Consequently, we employed the Kruskal-Wallis test. Post hoc analysis was performed using a Dunn test with Bonferroni corrections, comparing each treatment group with the control group. To compare changes in the same animals between LOC1 and LOC2, we used the paired Wilcoxon test. We performed these comparisons for groups that were statistically significant compared to the control on the LOC1 or LOC2.

All statistical calculations and graphs were done in the R Studio version 2023.06. with R version 4.2.1 [37]. The boxplot figures depict the median and interquartile range (IQR). The whiskers extend to the minimum and maximum values, excluding outliers that are defined by being outside of 1.5x IQR distance from the first quartile downwards or the third quartile upwards. For the Kruskal-Wallis test, we report the test statistic ($\chi2$), the degrees of freedom ($df$), and the corresponding p-value ($p$). For the Dunn test, we report the value of the test statistic ($Z$) and the p-value ($p$). Lastly, in the case of the paired Wilcoxon test, we provide the value of the test statistic ($V$) and its corresponding p-value ($p$).

# 3. Results

## 3.1 Locomotor stimulation and sensitization

We analyzed the distance moved after the initial and repeated treatments to evaluate the locomotor stimulation and sensitization (Fig 2). We performed the LOC1 with the first treatment dose on day one and the LOC2 with the seventh treatment dose on day 19.

Using the Kruskal-Wallis test, we observed a significant difference in the distance moved during the LOC1 with the first treatment dose ($\chi2 = 24.860$, $df = 6$, $p < 0.001$) and during the LOC2 with the final treatment dose ($\chi2 = 40.085$, $df = 6$, $p < 0.001$). Post-hoc analysis using the Dunn test with Bonferroni correction revealed that during LOC1, only S15 stimulated locomotion compared to saline ($Z = 2.912$, $p = 0.022$). During LOC2, R15 ($Z = 3.015$, $p = 0.008$), S15 ($Z = 5.036$, $p < 0.001$), RS15 ($Z = 3.170$, $p = 0.005$) and RS30 ($Z = 4.768$, $p < 0.001$) stimulated locomotion compared to saline. Lastly, we evaluated locomotor sensitization using paired Wilcoxon test on groups that were significant on LOC1 or LOC2 with the Dunn test. We observed a statistically significant increase in distance traveled during LOC2 compared to LOC1 for R15 ($V = 0$, $p = 0.016$), S15 ($V = 0$, $p = 0.008$), and RS15 ($V = 0$, $p = 0.016$), but not for RS30 ($V = 0$, $p = 0.062$).

Furthermore, we observed a distinct time course of changes in locomotion after administering S15 and RS30 during LOC1 (Fig 3). S15 led to a rapid increase in movement followed by a transient decrease and another increase. The administration of RS30 in LOC1 initially led to a cessation of locomotion, with locomotor stimulation appearing only after 40 minutes. During LOC2, we did not observe a transient decrease or cessation of movement for S15 and RS30.

When evaluating the time course of locomotor effects of treatment with ketamine and ketamine enantiomers, we also observed interindividual differences (S1 File).

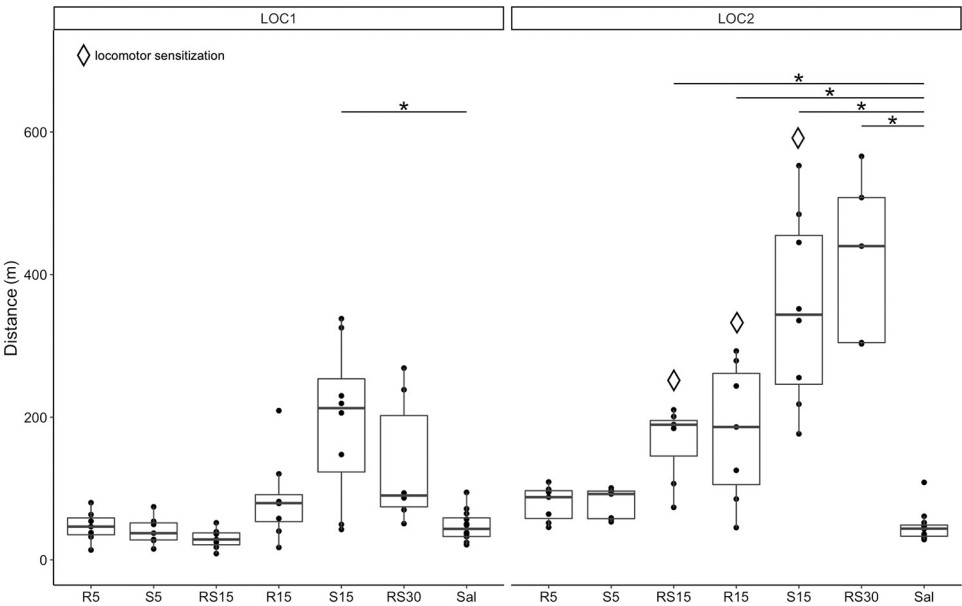

**Fig 2. Locomotor stimulation and sensitization after treatment with ketamine enantiomers and racemic ketamine.** Distance moved in 1 hour after treatment (y-axis): comparison of different treatment groups to control (x-axis) during LOC1 (left) and LOC2 (right). * indicates a statistically significant difference, $p < 0.05$. Diamond shape—significant increase in distance moved during LOC2 compared to LOC1 (locomotor sensitization), $p < 0.05$. R5–5 mg/kg of R-ketamine; S5–5 mg/kg of S-ketamine; RS15–15 mg/kg of racemic ketamine; R15–15 mg/kg of R-ketamine; S15–15 mg/kg of S-ketamine; RS30–30 mg/kg of racemic ketamine; Sal—saline.

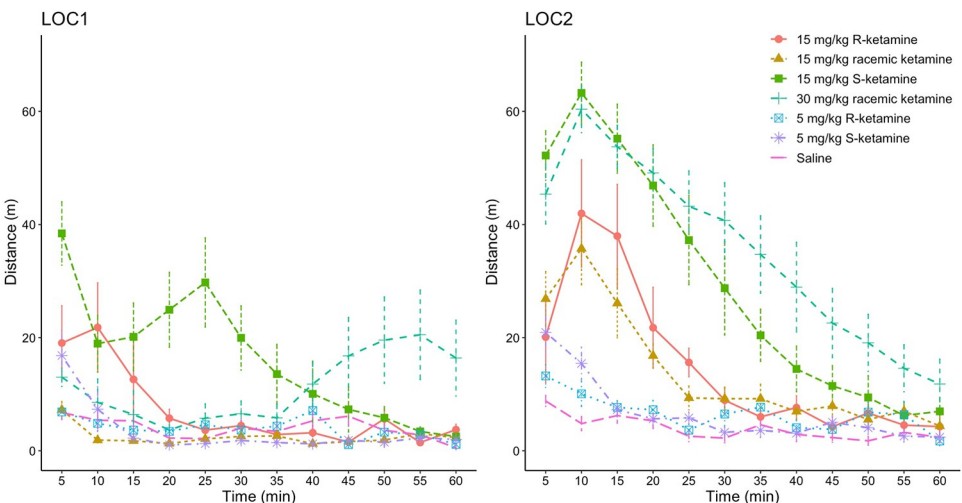

**Fig 3. Time course of locomotor effects for R-ketamine, S-ketamine, and racemic ketamine.** Each point represents a distance moved in 5 minutes (expressed in meters).

### 3.2 Ataxia

To evaluate ataxia after the initial and repeated treatment, we evaluated a relative number of falls in the first 15 minutes following treatment (Fig 4). For a RS30 group, we also analyzed the last 15 minutes of both LOC tests (RS30-L).

The Kruskal-Wallis test revealed significant differences between treatment groups in the relative number of falls during LOC1 ($\chi 2 = 39.392$, $df = 7$, $p < 0.001$) and LOC2 ($\chi 2 = 34.791$, $df = 7$, $p < 0.001$). Post hoc analysis using the Dunn test with Bonferroni correction showed a significantly higher relative number of falls in the first 15 minutes of the LOC1 for S15 ($Z = 2.554$, $p < 0.001$) and in the last 15 minutes of the LOC1 for RS30 (RS30-L; $Z = 4.188$, $p < 0.001$). During LOC2, the relative number of falls increased for S15 ($Z = 3.175$, $p = 0.011$), RS15 ($Z = 2.948$, $p = 0.022$) and RS30 ($Z = 4.448$, $p < 0.001$) compared to the control group.

To evaluate possible tolerance to the ataxic effects after repeated treatment, we compared the relative number of falls during LOC1 and LOC2. Our results show that on LOC2, S15 resulted in a lower relative number of falls than on LOC1 ($V = 28$, $p = 0.022$). This was not true for RS15 ($V = 3$, $p = 0.281$), RS30 ($V = 3$, $p = 0.312$), and RS30-L ($V = 15$, p = 0.063).

We also observed interindividual differences when evaluating the ataxic effects of treatment with ketamine (S1 File).

### 3.3 Natural behaviors

To evaluate changes in natural behaviors, we measured the duration of grooming (Fig 5) and rearing (Fig 6) during the first 15 minutes of LOC1 and LOC2.

The Kruskal-Wallis test was significant when analyzing the duration of grooming and rearing during the first 15 minutes of LOC1 (grooming: $\chi 2 = 37.313$, $df = 6$, $p < 0.001$, rearing: $\chi 2 = 32.736$, $df = 6$, $p < 0.001$) and LOC2 (grooming: $\chi 2 = 32.736$, $df = 6$, $p < 0.001$, rearing: $\chi 2 = 16.431$, $df = 6$, $p = 0.012$). Grooming duration was shorter in LOC1 and LOC2 for S5 (LOC1: $Z = 4.016$, $p < 0.001$; LOC2: $Z = 3.258$, $p = 0.007$), S15 (LOC1: $Z = 4.765$, $p < 0.001$; LOC2: $Z = 3.666$, $p = 0.001$), and RS30 (LOC1: $Z = 3.209$, $p = 0.008$; LOC2: $Z = 3.197$, $p = 0.008$), and only in LOC2 for RS15 ($Z = 3.705$, $p = 0.001$) according to Dunn test with Bonferroni correction. Furthermore, rearing duration was shorter in LOC1 for treatment groups

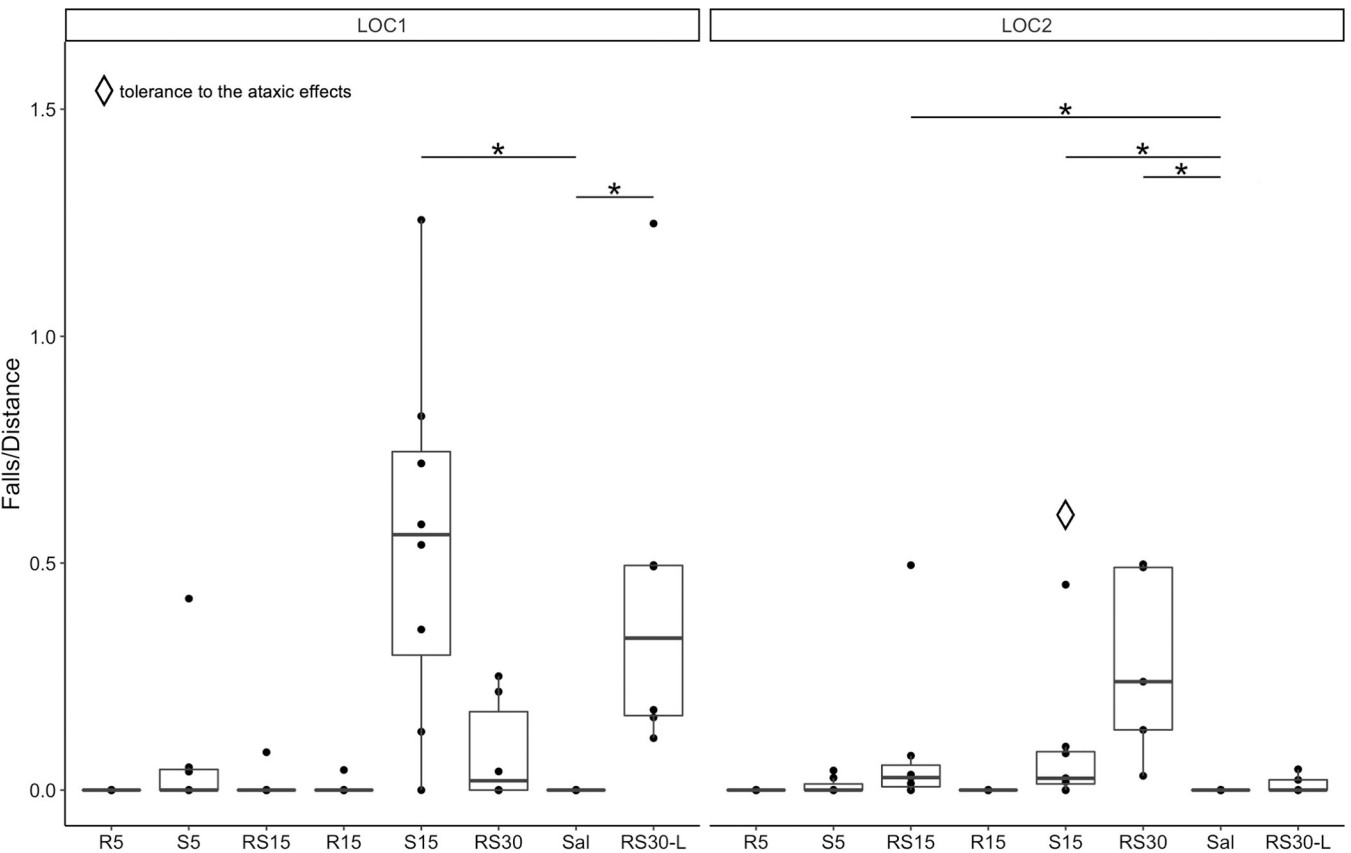

**Fig 4. Relative number of falls after initial and repeated treatment with R-ketamine, S-ketamine, and racemic ketamine.** Relative number of falls: comparing treatment groups with a control group during LOC1 (left) and LOC2 (right). * indicates a statistically significant difference, $p < 0.05$. Diamond shape—significant decrease in relative number of falls on LOC2 compared to LOC1 (tolerance to the ataxic effects), $p < 0.05$. R5–5 mg/kg of R-ketamine; S5–5 mg/kg of S-ketamine; RS15–15 mg/kg of racemic ketamine; R15–15 mg/kg of R-ketamine; S15–15 mg/kg of S-ketamine; RS30–30 mg/kg of racemic ketamine; Sal—saline; RS30-L—last 15 minutes of 30 mg/kg of racemic ketamine.

S5 ($Z = 2.826$, $p = 0.028$) and S15 ($Z = 3.541$, $p = 0.002$), and in LOC2 for S15 ($Z = 3.133$, $p = 0.010$), according to Dunn test with Bonferroni correction.

## 4. Discussion

This study investigated the differential behavioral effects of repeated treatment with ketamine enantiomers and racemic ketamine in subanesthetic doses. We focused on locomotor stimulation, locomotor sensitization, ataxia, and natural behaviors (grooming and rearing).

Administering S15, but not S5, resulted in locomotor stimulation, locomotor sensitization, and ataxia. This finding is consistent with previous research on S-ketamine showing locomotor stimulation after 10 mg/kg i.p. [8, 10, 14], 20 mg/kg i.p. [8, 10], 30 mg/kg i.p. [14], and 10, 20 and 40 mg/kg via intranasal route [11], but not after 5 mg/kg i.p. [8, 10]. Locomotor sensitization was shown after repeated treatment with 20 mg/kg of S-ketamine i.p. [8] and ataxia after 20 mg/kg and 50 mg/kg i.p. [12]. Our study additionally showed that S5 and S15 reduces the duration of grooming and rearing. No other study compared the changes in natural behaviors after treatment with S-ketamine.

Contrary to S-ketamine, R15 did not stimulate locomotion after the first treatment. After repeated treatment, however, locomotor stimulation and sensitization occurred. While some studies showed that R-ketamine does not stimulate locomotion at 5, 10 and 20 mg/kg i.p. [10]

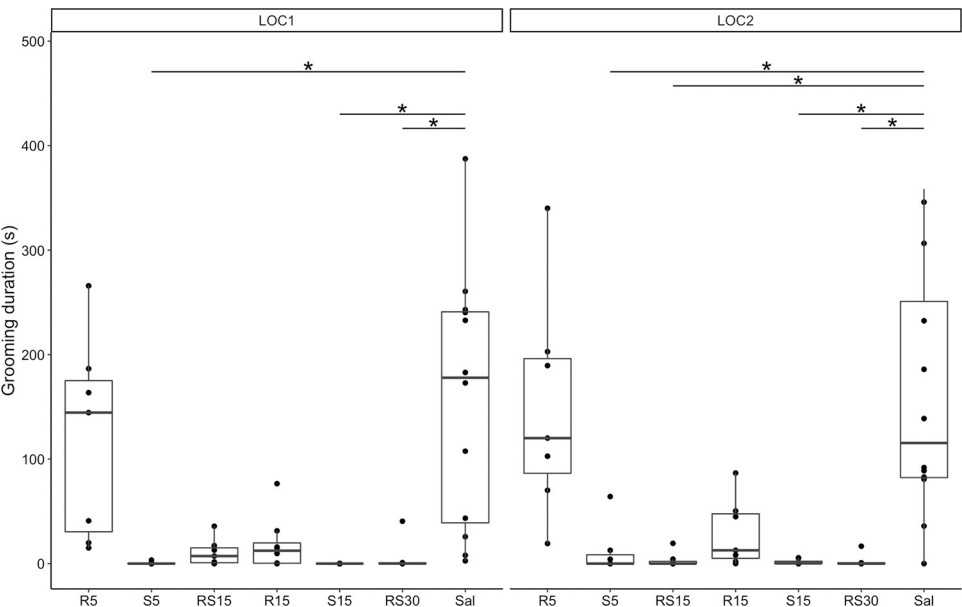

**Fig 5. Grooming duration after initial and repeated treatment with R-ketamine, S-ketamine, and racemic ketamine.** Grooming duration: comparing treatment groups with a control group during LOC1 (left) and LOC2 (right). * indicates a statistically significant difference, $p < 0{,}05$. R5–5 mg/kg of R-ketamine; S5–5 mg/kg of S-ketamine; RS15–15 mg/kg of racemic ketamine; R15–15 mg/kg of R-ketamine; S15–15 mg/kg of S-ketamine; RS30–30 mg/kg of racemic ketamine; Sal—saline.

and 10, 20 and 40 mg/kg via intranasal route [11], others showed that R-ketamine does stimulate locomotion at 20 mg/kg i.p. [8], and 10 and 30 mg/kg i.p. [14]. Contrary to our study, Bonaventura et al. [8] also found that R-ketamine does not lead to locomotor sensitization. The

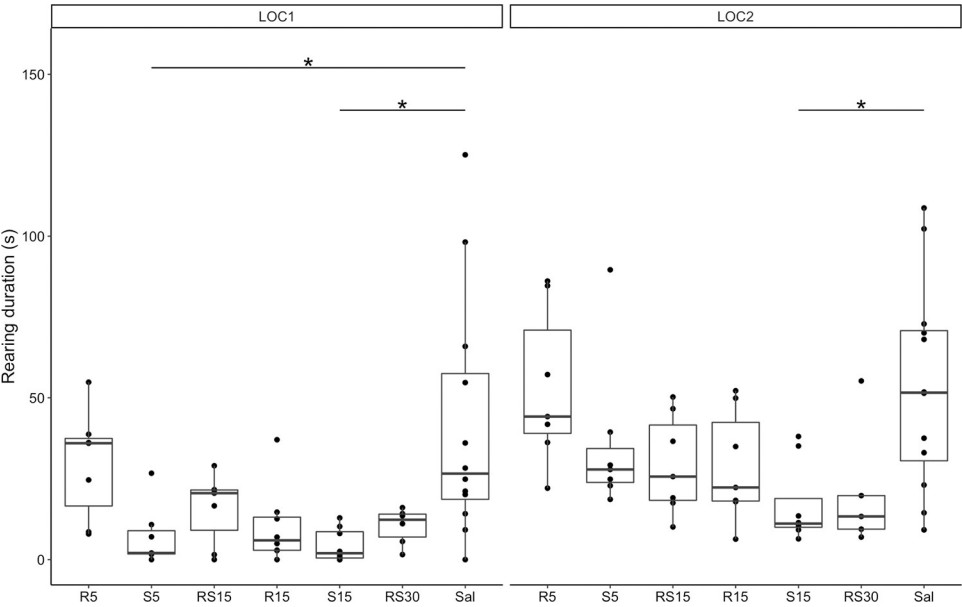

**Fig 6. Rearing duration after initial and repeated treatment with R-ketamine, S-ketamine, and racemic ketamine.** Rearing duration: comparing treatment groups with a control group during LOC1 (left) and LOC2 (right). * indicates a statistically significant difference, $p < 0{,}05$. R5–5 mg/kg of R-ketamine; S5–5 mg/kg of S-ketamine; RS15–15 mg/kg of racemic ketamine; R15–15 mg/kg of R-ketamine; S15–15 mg/kg of S-ketamine; RS30–30 mg/kg of racemic ketamine; Sal—saline.

discrepancy between the studies may be attributed to variations in experimental species and treatment protocols. Bonaventura et al. used mice while we used rats, and they administered three consecutive doses of 20 mg/kg i.p., one per day. In contrast, we administered seven doses of 15 mg/kg with a three-day interval between them using the s.c. route of administration. Lastly, in our study, R5 did not stimulate locomotion and resulted in locomotor sensitization, and neither R5 nor R15 led to ataxia or change in natural behaviors. No other studies evaluated natural behaviors after R-ketamine. Only one study evaluated ataxic behaviors after ketamine enantiomers and found that R-ketamine causes ataxia at 50 mg/kg i.p., which was weaker than that of S-ketamine at 20 and 50 mg/kg i.p. [12].

Besides ketamine enantiomers, we also evaluated the effects of racemic ketamine. The first dose of RS15 and RS30 did not stimulate locomotion. However, RS30 did result in ataxia. With repeated treatment, both doses stimulated locomotion and resulted in ataxia. RS15 also led to locomotor sensitization. Notably, the first dose of RS30 and the last dose of RS15 and RS30 reduced the duration of natural behaviors (grooming).

Strong et al. [18] demonstrated that racemic ketamine stimulates locomotion at 2.5 and 5 mg/kg i.p. and leads to locomotor sensitization at 5 mg/kg i.p. Contrary, Trujillo et al. [22] did not find these effects at 5 mg/kg i.p. Locomotor stimulation occurred at 10 and 20 mg/kg i.p., and locomotor sensitization only at 20 mg/kg i.p. Hetzler and Swain-Wautlet [21] reported locomotor stimulation after 50 and 100, but not 1 and 10 mg/kg i.p.

Furthermore, it has been observed that racemic ketamine can cause ataxia at 2 and 5 mg/kg when administered via intravenous bolus (i.v.b.) [28, 29]. Similar effects have been reported with 10 mg/kg [21], 25 mg/kg [25, 26], 50 mg/kg, and 100 mg/kg [21] via i.p. as well as 20 mg/kg/h via intravenous infusion (i.v.i.) [28]. Existing literature also shows that racemic ketamine reduces natural behaviors at 2 and 5 mg/kg i.v.b. [28, 29], 5, 10, and 20 mg/kg/h i.v.i [28] and 10, 20, and 40 mg/kg i.p. [30].

Taken together, racemic ketamine stimulates locomotion, results in locomotor sensitization and ataxia, and decreases the expression of natural behaviors. However, doses to trigger these behavioral effects differ between the studies. The discrepancy in doses required to induce behavioral effects of ketamine may partially stem from the route of administration; i.v.b. achieves higher ketamine exposure in comparison with i.v.i. and s.c. [31], which may lead to more potent effects at lower doses. This may explain why ataxia was found after 2 and 5 mg/kg i.v.b. [28, 29] but not at 5 and 10 mg/kg/h i.v.i. [28], and after S5 and R5 s.c. in our study.

Unfortunately, no studies directly compared s.c ketamine with the most commonly used i. p. administration of ketamine. However, the ketamine/xylazine mixture was proven to have comparable anesthetic effectiveness between i.p. and s.c. [38]. A recent review [39] found that the i.p. route allows quicker and more complete absorption of substances than the s.c. route. Additionally, it highlighted that i.p. administration leads to faster drug metabolism due to a first-pass effect akin to oral administration. Further research is necessary to explore the implications of i.p. and s.c. administration routes for ketamine in preclinical studies, especially because s.c. ketamine produces exposure levels similar to the i.v.i., which is commonly used in clinical settings [31]. S.c. administration of ketamine also showed clinical relevance [32–34] while being safer than i.v. route [32, 34].

Despite utilizing the same route of administration, the behavioral effects of racemic ketamine can still vary. Strong et al. [18] found locomotor stimulation after 2.5 and 5 mg/kg i.p. and locomotor sensitization after 5 mg/kg i.p. On the contrary, Trujillo et al. [22] did not find these effects at 5 mg/kg but locomotor stimulation at 10 and 20 mg/kg i.p. and locomotor sensitization at 20 mg/kg i.p. Both studies were conducted with male Sprague Dawley rats (Strong et al. also used females) of similar weights and under comparable conditions, including testing during the light phase and using similar apparatuses. The discrepancy between the studies

highlights the complex interplay of factors influencing the behavioral effects of ketamine, extending beyond dosage and route of administration. The reasons for these differences remain unknown, suggesting that variables such as experimental conditions, individual animal responses, or methodological differences might significantly impact outcomes.

In our study, we also observed a distinct time course of locomotion after initial administration of S15 and RS30 (Fig 3, LOC1). It was characterized by an initial decrease in movement followed by locomotor stimulation. Hetzler and Swain Wautlet [21] reported a similar time course of behavioral effects after 50 mg/kg and 100 mg/kg i.p. of racemic ketamine. We attribute these movement patterns to ataxic effects predominating over the locomotor effects of ketamine. Initially, the ataxic symptoms are strong enough to inhibit or complicate movement. As these effects diminish, locomotor stimulation becomes increasingly evident. However, Yang et al. [10] did not report such a locomotion profile after 10 and 20 mg/kg of S-ketamine. This variance could be attributed to differences in species (mice versus rats) and the route of administration (i.p. versus s.c.). Hetzler and Swain Wautlet additionally suggest that the type and size of the arena might influence locomotor stimulation patterns following ketamine treatment [21].

The characteristic drop in movement observed after the initial treatment was no longer present after repeated treatment (Fig 3, LOC2). We interpret the change in the time-course characteristics as indicative of the development of tolerance to the ataxic effects of ketamine. We observed a significant reduction of ataxic behaviors after repeated treatment with S15 (Fig 4). In contrast, Uchihashi et al. [26] found no evidence of tolerance to the ataxic effects of racemic ketamine following repeated treatment with 25 mg/kg using i.p. The discrepancy between the two studies may be attributed to the differences in the behavioral paradigms. While they employed the rotarod test, we utilized manual observations of behavior, allowing for a more detailed assessment.

Similarly to our study, tolerance to repeated treatment with ketamine was found regarding its anesthetic efficacy [40–43] and psychotomimetic effects [44]. The tolerance reported in our study may indicate that repeated treatment of patients suffering from depression with ketamine or ketamine enantiomers would progressively result in fewer side effects. However, tolerance could also develop to antidepressant effects. We are not aware of any clinical studies on possible tolerance to antidepressant effects, except a case report indicating tolerance after unregulated, self-medicated ketamine use [45]. Preclinical studies on long-term antidepressant efficacy are scarce and with inconsistent results [46, 47].

We show that repeated treatment with R-ketamine and S-ketamine causes locomotor stimulation and sensitization. However, S-ketamine has stronger stimulating effects, causes ataxia, and decreases natural behaviors, while R-ketamine does not. Similarly, other studies found that S-ketamine has a stronger behavioral effect than R-ketamine. S-ketamine leads to conditioned place preference and altered prepulse inhibition at 10 and 20 mg/kg i.p. [10], reliable self-administration [8], it cross-sensitizes to methamphetamine at 10 mg/kg i.p. [9], and results in ataxia and stereotypy at 20 and 50 mg/kg i.p. [12]. R-ketamine leads to some of these behaviors, but at higher doses: conditioned place preference at 40 mg/kg i.p. [48], altered prepulse inhibition at 90 mg/kg i.p. [48], and ataxia and stereotypy at 50 mg/kg i.p., which were weaker than that of S-ketamine at 20 and 50 mg/kg i.p. [12].

Our study confirms that S-ketamine has stronger behavioral effects than R-ketamine. It also shows that S-ketamine has stronger behavioral effects than racemic ketamine because S-ketamine causes locomotor stimulation and ataxia already with the initial dose in LOC1 and has stronger stimulating effects at 15 mg/kg s.c. in LOC2. Additionally, our research highlights that racemic ketamine exerts stronger behavioral effects than R-ketamine, given that racemic ketamine leads to ataxia and a reduction in natural behaviors at 15 mg/kg s.c., an outcome not

observed with R-ketamine. Similarly, Chang et al. [11] showed that S-ketamine had stronger effects on locomotor stimulation, prepulse inhibition, and conditioned place preference compared to racemic ketamine and R-ketamine, while racemic ketamine had stronger effects compared to R-ketamine.

Since S-ketamine causes stronger behavioral effects, which are usually regarded as unwanted effects, we might reason that it is less desired as a therapeutic tool. Nevertheless, when interpreting the effects of ketamine enantiomers as desirable or undesirable, it is crucial to consider the therapeutic goal and efficacy. S-ketamine has higher efficacy and fewer side effects than R-ketamine when used as an analgesic [20, 49] and anesthetic [20, 50]. However, when ketamine is used as an antidepressant, things are less clear. There is a lack of clinical comparison of antidepressant potential between ketamine enantiomers. Recent systematic reviews report that S-ketamine and racemic ketamine reduce symptoms of depression and that racemic ketamine might be more efficacious [51, 52]. Unfortunately, R-ketamine has been poorly researched, with only two studies that yielded contradictory findings [53, 54]. On the other hand, preclinical evidence suggests R-ketamine leads to higher antidepressant efficacy [10, 11, 13] with fewer unwanted effects [8, 9]. However, a recent preclinical study reported the antidepressant effect of S-ketamine but not of R-ketamine [55], possibly indicating that the model of depression or treatment protocol also plays a role in treatment efficacy. Future studies are needed to conclude the antidepressant efficacy of both enantiomers [56].

Lastly, there are several limitations to consider when interpreting the results of our study. We used only males to avoid hormonal fluctuations in the long-term protocol. However, it is worth noting that ketamine sensitivity and efficacy are slightly different between male and female animals and that hormonal fluctuations in females affect sensitivity to ketamine [57–59]. We had a reversed light cycle, meaning that we performed behavioral tests in light during the night phase, which might have disrupted the circadian rhythm of the animals and resulted in behavioral changes [60]. Our study was performed mainly by a male researcher. A female researcher helped with s.c. administration; however, she did not perform behavioral tests. It is well known that the sex of the researcher may influence the outcomes of behavioral tests and must be taken into account when interpreting the results [61, 62].

## 5. Conclusion

We found that the first dose of S15 stimulated locomotion and caused ataxia during LOC1. RS30 also caused ataxia during LOC1. After repeated treatment, R15, S15, RS15, and RS30 stimulated locomotion, and R15, S15, and RS15 resulted in locomotor sensitization. Repeated treatment with S15, RS15, and RS30 also resulted in ataxia. Furthermore, repeated treatment with S15 led to the development of tolerance to the ataxic effects. Lastly, RS15, RS30, S5, and S15 decreased the duration of natural behaviors, while R5 and R15 did not. Our findings are significant as this is the first study to report locomotor sensitization with R-ketamine, development of tolerance to the ataxic effects of S-ketamine, and distinct impacts of ketamine enantiomers on natural behaviors. In summary, our results provide further evidence that S-ketamine exerts more pronounced behavioral effects compared to R-ketamine.

## Supporting information

**S1 File.**
(DOCX)

## Acknowledgments

The authors express their gratitude to Tina Košuta for her valuable assistance with the statistical analysis.

## Author Contributions

**Conceptualization:** Kristian Elersič, Marko Živin, Maja Zorović.

**Data curation:** Kristian Elersič, Anamarija Banjac.

**Formal analysis:** Kristian Elersič, Anamarija Banjac.

**Funding acquisition:** Marko Živin.

**Investigation:** Kristian Elersič, Maja Zorović.

**Methodology:** Kristian Elersič, Anamarija Banjac, Marko Živin, Maja Zorović.

**Project administration:** Marko Živin.

**Resources:** Marko Živin.

**Software:** Kristian Elersič.

**Supervision:** Marko Živin.

**Validation:** Marko Živin, Maja Zorović.

**Visualization:** Kristian Elersič.

**Writing – original draft:** Kristian Elersič.

**Writing – review & editing:** Kristian Elersič, Anamarija Banjac, Marko Živin, Maja Zorović.

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
