## [Decision Letter · Decision Letter 0]

2 Oct 2023

PONE-D-23-30728Behavioral sensitization and tolerance induced by repeated treatment with ketamine enantiomers in male Wistar ratsPLOS ONE

Dear Dr. Elersič,

Thank you for submitting your manuscript to PLOS ONE. After careful consideration, we feel that it has merit but does not fully meet PLOS ONE’s publication criteria as it currently stands. Therefore, we invite you to submit a revised version of the manuscript that addresses the points raised during the review process.

 Although the data of your manuscript are interesting, the following concerns from two reviewers should be addressed.

We look forward to receiving your revised manuscript.

Kind regards,

Kenji Hashimoto, PhD

Section Editor

PLOS ONE

Journal Requirements:

Reviewers' comments:

Reviewer's Responses to Questions

**Comments to the Author**

1. Is the manuscript technically sound, and do the data support the conclusions?

Reviewer #1: Yes

Reviewer #2: Partly

2. Has the statistical analysis been performed appropriately and rigorously? 

Reviewer #1: Yes

Reviewer #2: Yes

3. Have the authors made all data underlying the findings in their manuscript fully available?

Reviewer #1: Yes

Reviewer #2: Yes

4. Is the manuscript presented in an intelligible fashion and written in standard English?

Reviewer #1: Yes

Reviewer #2: Yes

5. Review Comments to the Author

Reviewer #1: The results of this study showed that repeated treatment with S-ketamine cause locmotor sensitization and tolerance to the ataxic effects, although R-ketamine also has simiar behavior but it is not observed before repeated treatment. The study could be accepted after major revision:

1) racemic ketamine could be included in the study, I think addtional experiemnts of racemic ketamine will strengthen the conclusion of the study and make it more convincing.

2) The authors need to show the evidence that 15 mg/kg is a suitable dosage choice. So this study could add a dose-dependent prelimimnary study comparing 5, 10 and 15 mg/kg.

3) why the authors did not adopt two administration of ketamine per week, instead of every 3 days, this protocol has been accepted for depression treatment by ketamine clinically.

4) I could not find the figure legends for the study.

Reviewer #2: In this study, authors compared abuse potential and psychotomimetic potential of two ketamine enantiomers, S-ketamine and R-ketamine by means of locomotor sensitization and stereotyped behavior. Based on the results, authors concluded that S-ketamine has higher abuse potential and psychotomimetic potential than R-ketamine. Given that there are many reports demonstrating that R-ketamine has more potent antidepressant-like effects than S-ketamine, while that R-ketamine is much safer than S-ketamine, the objective of the study is important and timely. Although there are several studies which compared side effect liabilities of S-ketamine and R-ketamine, this is a nice addition to show differences of these enantiomers. However, there are several concerns on interpretation of the results, and experimental procedures. The followings are comments to be considered.

1. Authors used locomotor activity after acute treatment and potential to induce locomotor sensitization after repeated treatment. Behavioral sensitization is partly regarded as abuse potential, but this phenomenon is not equal to abuse potential. For example, reference #17 measured molecular changes relating abuse potential in the nucleus accumbens which are accompanied with behavioral sensitization to relate this behavioral changes with abuse potential. Please be very careful to clearly mention abuse potential by only the results of behavioral sensitization.

Moreover, it is premature to say that increase in locomotor activity after acute administration is indicative of abuse potential.

Likewise, both ataxia and stereotyped behavior do not necessarily reflect psychotomimetic/dissociative symptoms, because these behavioral abnormalities may be induced by several factors including impaired motor coordination.

2. Please justify 15 mpk SC dose that authors used for this study. Although authors claimed that doses were selected based on the previous reports (and cited references accordingly), intraperitoneal administration was used in the previous reports. In contrast, authors used subcutaneous administration, which is expected to provide higher exposure of both enantiomers, because ketamine is rapidly and extensively metabolized before entering circulation in case of IP administration but not SC administration. Indeed, S-ketamine induced ataxia at 15 mpk SC, a condition which cannot be used to evaluate antidepressant-like effects in FST. Please mention possible differences in exposure between the present study and the previous ones. Moreover, please discuss differences in exposure of ketamine enantiomer between IP administration and SC administration. Without discussion on exposure level, it is difficult to compare doses for abuse potential with ones for antidepressant-like effects.

3. Accordingly, authors’ conclusion that R-ketamine has abuse potential is not appropriate. Zanos et al. suggested abuse potential at higher dose than dose showing antidepressant-like effects. Therefore, authors need to clarify that dose used in the present study may be higher than one showing antidepressant effects.

4. Figure 2. Please show time course of locomotion as well. Therefore, authors may incorporate Figure 4A into Figure 2. Instead, it is not necessary to show both A and B in Figure 1, and individual data in Figure 4 can be transferred to supplementary data. Because ataxia and behavioral abnormalities were observed in S-ketamine dose, it is not meaningful to compare peak time in locomotor increase between R-ketamine and S-ketamine.

5. Regarding description on more robust antidepressant-like effects of R-ketamine than S-ketamine, the followings are representative reports on this finding. Authors should cite these reports.

Zhang et al., Pharmacol Biochem Behav. 2014 Jan;116:137-41.

Yang et al., Transl Psychiatry. 2015 Sep 1;5(9):e632.

6. There are many citations which are not appropriate. For example, abuse potential of ketamine is not described in reference #10, and reference #12 did not mention differential side effects between S-ketamine and R-ketamine. Please carefully confirm contents of all of citations, and correct them accordingly.

6. PLOS authors have the option to publish the peer review history of their article (what does this mean?). If published, this will include your full peer review and any attached files.

Reviewer #1: No

Reviewer #2: No

---

## [Author Response · Author response to Decision Letter 0]

6 Feb 2024

Dear Kenji Hashimoto, PhD,

Thank you for providing detailed feedback and expressing your concerns regarding our manuscript. We have thoroughly revised the manuscript in accordance with the valuable suggestions of the reviewers.

Enclosed, please find our point-by-point responses to each of the specific comments. We hope that these revisions adequately address the concerns raised.

Should there be any additional modifications or information required, please do not hesitate to contact us. We are committed to ensuring the manuscript meets your publication's standards.

Thank you for your attention to this matter.

Best regards,

Kristian Elersič, Anamarija Banjac, Marko Živin, Maja Zorović

Reviewer #1

1) 

We have incorporated racemic ketamine at doses of 15 mg/kg and 30 mg/kg into our study protocol. This inclusion has enabled us to validate that S-ketamine exhibits more pronounced behavioral effects than racemic ketamine and R-ketamine. Furthermore, we found that racemic ketamine has more pronounced behavioral effects than R-ketamine. 

2) 

To assess the impact of lower doses, we included 5 mg/kg of ketamine enantiomers in our study. This is the dose at which racemic ketamine still exhibited antidepressant effects when applied subcutaneously [1]. We did not include the 10 mg/kg dosage of ketamine enantiomers in our revision, as it is already being examined in a separate ongoing research project. This concurrent study involves older animals and compares the effects of ketamine enantiomers across Wistar and Wistar-Kyoto lines, utilizing a slightly modified behavioral protocol.

3) 

We acknowledge the concern regarding the frequency of treatments and its alignment with clinical protocols. While administering treatments twice weekly would more closely resemble a clinical practice, we deliberately chose a treatment schedule every third day. This approach ensures a consistent interval between treatments, thereby reducing variability in the treatment protocol. We aimed to minimize the potential variables that could confound the study's outcomes.

4) 

The legend has been incorporated into the Fig 3. In the remaining figures, we have revised and employed abbreviations consistent with those used throughout the text. Furthermore, a detailed explanation of the figure abbreviations has been included in the corresponding figure captions. Please advise if it is necessary to include legends directly within the figures; however, we aimed to maintain a streamlined appearance by avoiding additional clutter.

Reviewer #2

1. 

Thank you for the raised concern. We have corrected the manuscript accordingly. In the revised manuscript, we are discussing our results only as behaviors that they are (locomotor stimulation, sensitization, ataxia, tolerance, and natural behaviors) and not as abuse and psychotomimetic potential. However, we briefly state that selected behaviors are frequently used as preclinical markers for unwanted side effects. 

2. 

We have additionally incorporated a lower dose of 5 mg/kg of ketamine enantiomers into our study, as this dosage has been previously demonstrated to exert an antidepressant effect with racemic ketamine when administered subcutaneously [1]. We have also included a discussion regarding the subcutaneous route of administration. 

3. 

We have revised our conclusion in alignment with the feedback received. We concur that a dosage of 15 mg/kg for R-ketamine may be excessively high for achieving antidepressant effects, particularly considering that 5 mg/kg of racemic ketamine, administered subcutaneously, has demonstrated antidepressant efficacy [1]. Unfortunately, no studies have explored the subcutaneous route of administration for comparing ketamine enantiomers. To draw definitive conclusions, future studies are warranted to assess the antidepressant efficacy of ketamine enantiomers using the subcutaneous route.

4. 

We have corrected the figures accordingly. 

5. 

We are grateful for the references provided. They have been incorporated into our manuscript.

6.

We sincerely apologize for the errors in our referencing. We have thoroughly reviewed and verified the content of each citation and made the necessary corrections. 

[1] C.W. McDonnell, F. Dunphy-Doherty, J. Rouine, M. Bianchi, N. Upton, E. Sokolowska, J.A. Prenderville, The Antidepressant-Like Effects of a Clinically Relevant Dose of Ketamine Are Accompanied by Biphasic Alterations in Working Memory in the Wistar Kyoto Rat Model of Depression, Front. Psychiatry 11 (2021). https://www.frontiersin.org/article/10.3389/fpsyt.2020.599588 (accessed February 10, 2022).

---

## [Decision Letter · Decision Letter 1]

9 Feb 2024

Behavioral sensitization and tolerance induced by repeated treatment with ketamine enantiomers in male Wistar rats

PONE-D-23-30728R1

Dear Dr. Elersič,

We’re pleased to inform you that your manuscript has been judged scientifically suitable for publication and will be formally accepted for publication once it meets all outstanding technical requirements.

Kind regards,

Kenji Hashimoto, PhD

Section Editor

PLOS ONE

Additional Editor Comments (optional):

Reviewers' comments:

Reviewer's Responses to Questions

**Comments to the Author**

1. If the authors have adequately addressed your comments raised in a previous round of review and you feel that this manuscript is now acceptable for publication, you may indicate that here to bypass the “Comments to the Author” section, enter your conflict of interest statement in the “Confidential to Editor” section, and submit your "Accept" recommendation.

Reviewer #1: All comments have been addressed

Reviewer #2: All comments have been addressed

2. Is the manuscript technically sound, and do the data support the conclusions?

Reviewer #1: Yes

Reviewer #2: Yes

3. Has the statistical analysis been performed appropriately and rigorously? 

Reviewer #1: Yes

Reviewer #2: Yes

4. Have the authors made all data underlying the findings in their manuscript fully available?

Reviewer #1: Yes

Reviewer #2: Yes

5. Is the manuscript presented in an intelligible fashion and written in standard English?

Reviewer #1: Yes

Reviewer #2: Yes

6. Review Comments to the Author

Reviewer #1: This is a timely study comparing the ketamine enantiomers. The authors have addressed all the comments. No addtional comments. This revised manuscript could be accepted in Plos One.

Reviewer #2: Authors responded adequately to all of my comments, conducted additional experiments as requested, and revised the manuscript accordingly. I appreciate all of authors' efforts to improve the manuscript. I have no further comments.

7. PLOS authors have the option to publish the peer review history of their article (what does this mean?). If published, this will include your full peer review and any attached files.

Reviewer #1: No

Reviewer #2: No

---

## [Editor Report · Acceptance letter]

22 Feb 2024

PONE-D-23-30728R1 

PLOS ONE

Dear Dr. Elersič, 

I'm pleased to inform you that your manuscript has been deemed suitable for publication in PLOS ONE. Congratulations! Your manuscript is now being handed over to our production team.

Kind regards, 

on behalf of

Prof. Kenji Hashimoto 

Section Editor

PLOS ONE